# In Vitro Shoot Cultures of *Micromeria graeca*: Micropropagation and Evaluation of Methanolic Extracts for Anticancer and Antimicrobial Activity

**DOI:** 10.3390/plants14162592

**Published:** 2025-08-20

**Authors:** Branka Uzelac, Mirjana Janjanin, Dijana Krstić-Milošević, Gordana Tovilović-Kovačević, Đurđica Ignjatović, Tatjana Mihajilov-Krstev, Dragana Stojičić

**Affiliations:** 1Department of Plant Physiology, Institute for Biological Research “Siniša Stanković”—National Institute of the Republic of Serbia, University of Belgrade, Bulevar despota Stefana 142, 11108 Belgrade, Serbia; mirjana.janjanin@ibiss.bg.ac.rs (M.J.); dijana@ibiss.bg.ac.rs (D.K.-M.); 2Department of Biochemistry, Institute for Biological Research “Siniša Stanković”—National Institute of the Republic of Serbia, University of Belgrade, Bulevar despota Stefana 142, 11108 Belgrade, Serbia; tovilovicg@ibiss.bg.ac.rs (G.T.-K.); djurdjica@ibiss.bg.ac.rs (Đ.I.); 3Department of Biology and Ecology, Faculty of Sciences and Mathematics, University of Niš, Višegradska 33, 18000 Niš, Serbia; tatjana.mihajilov-krstev@pmf.edu.rs (T.M.-K.); dragana.stojicic@pmf.edu.rs (D.S.)

**Keywords:** axillary bud induction, Greek savory, activated charcoal, phenolic acids, anti-hepatoma activity

## Abstract

*Micromeria graeca* (L.) Benth. ex Rchb. (Lamiaceae) is a promising medicinal plant valued for its antioxidant, anti-hyperglycemic, anti-hypertensive, antimicrobial, and anti-aflatoxigenic properties. It is rich in phenolic and flavonoid compounds, supporting its traditional use for digestive, respiratory, cardiovascular, and dermatological conditions. Plant tissue culture facilitates controlled in vitro propagation to study plant growth and bioactive properties. The effects of activated charcoal and varying subculture intervals on multiplication and biomass production in *M. graeca* shoot cultures were investigated. The phenolic composition of methanolic extracts from in vitro-grown plants was characterized using high-performance liquid chromatography (HPLC), identifying rosmarinic, caffeic, and syringic acids as the primary phenolic compounds. Antimicrobial activity against selected microbial strains was evaluated using a micro-well dilution assay. Anticancer activity of selected extracts was assessed in human hepatocellular carcinoma cell line HepG2, with flow cytometry (Annexin-V/PI staining) used to analyze cell death mechanisms, and compared to pure rosmarinic acid (RA). Activated charcoal showed no beneficial effects on multiplication or biomass production, but significantly increased phenolic acid content (up to 4-fold). RA dominated the phenolic profiles, with other phenolic acids present in lower amounts. Methanolic extracts exhibited negligible antimicrobial activity compared to reference antibiotics and fungicide. Extracts from 4-week-old shoot cultures displayed modest anti-hepatoma activity (IC50 values of CV assay ranging from 193 to 274 µg mL^−1^), inducing HepG2 cell apoptosis via oxidative stress, independent of RA. Our results suggest that the metabolic output of *M. graeca* shoot cultures and consequently their biological activity can be modulated by varying in vitro culture conditions. These findings underscore the potential of their methanolic extracts for biotechnological production and therapeutic applications.

## 1. Introduction

The genus *Micromeria* Benth. (Lamiaceae) comprises perennial herbs, sub-shrubs and shrubs, rarely annual herbs, distributed throughout the temperate belt. This genus currently includes about 54 species [1], with ongoing taxonomic revisions [2,3] and new discoveries [4] potentially altering this number as molecular and morphological studies refine species boundaries. The distribution range of *Micromeria* species extends from the Himalayan region to the Macaronesia and from the Mediterranean to South Africa and Madagascar [5]. Reaching from the sea level to more than 2000 m in the Mediterranean and Macaronesia, *Micromeria* species mostly occur in moderately dry areas with a rather long dry season [1].

*Micromeria graeca* (L.) Benth. ex Rchb. (syn. *Satureja graeca* L.), commonly known as Greek savory or hyssop, is a perennial herb widely distributed in the Mediterranean area [6,7]. Like many members of the genera *Satureja* and *Micromeria*, *M. graeca* has been applied in traditional medicine across various cultures for its diverse therapeutic properties, attributed to its rich phytochemical profile, and its dried aerial parts are used as a condiment. In Mediterranean regions, *M. graeca* has been employed in folk medicine for centuries to treat digestive disorders [8], respiratory conditions [9,10,11], cardiovascular disorders [12], diabetes [13], skin infections [14], pain, and inflammation [8,9]. Scientific studies have validated many of these traditional uses, confirming its pharmacological properties such as antioxidant and antimicrobial activity [15,16,17,18], anti-hyperglycemic and anti-hyperlipidemic effects [13], anti-hypertensive and cardiovascular effects [12], and anti-enzymatic effects, such as tyrosinase, acetylcholinesterase, and α-amylase inhibition [18].

Growing demand for medicinal plants, valued for their potent biological activities, fuels overharvesting of wild populations, necessitating in vitro culture to safeguard genetic diversity and habitats [19]. Developing micropropagation protocols is essential for in vitro plant studies and supports long-term conservation efforts [20]. In vitro cultivation is a viable alternative and offers the opportunity to overcome the problems that are inherent in herbal extracts (genetic and phenotypic variability, extract variability and instability, toxic components and contaminants) as well as to manipulate phenotypic variation in bioactive compounds [21]. Biotechnology of medicinal plants is an innovative technology that provides a range of possibilities to obtain new sources of health-promoting compounds, including those from rare and protected plant species. In vitro culture techniques enable the production of consistent, renewable biomass from medicinal plant taxa, aiming to enhance the production of desirable secondary metabolites [22]. In plant cell and organ cultures, different chemical and physical factors can influence the biomass accumulation and synthesis of secondary metabolites [23,24,25]. Activated charcoal (AC) is often used in plant tissue culture to improve plant growth and development [26,27,28,29,30,31]. In addition to promoting morphogenesis, AC has been demonstrated to influence secondary metabolite production [31,32].

Biotechnologically derived extracts offer a controlled platform to enhance the production of health-promoting compounds and investigate their therapeutic potential [22]. Hepatocellular carcinoma is a major global health challenge, with the human hepatocellular carcinoma cell line (HepG2) widely used to evaluate the anticancer potential of natural compounds [33]. The phenolic compounds in *Micromeria* species, particularly rosmarinic acid, are promising candidates for anticancer activity [33,34,35], yet studies on extracts from in vitro cultured *M. graeca* are lacking. Although Badisa et al. [36] explored the cytotoxic effects of wild-growing *M. graeca*, this study examines in vitro cultured plants to assess how micropropagation conditions may influence phenolic metabolite profiles and their bioactivity. Similarly, the antimicrobial properties of wild-growing *M. graeca* plants, linked to their phenolic content and traditional use for infections, are documented [15,16,17], but evidence for in vitro-derived *M. graeca* extracts remains limited.

This study aimed to evaluate the effects of micropropagation conditions of *M. graeca* on potential bioactive compound production. We assessed the impact of activated charcoal and varying subculture durations on multiplication and biomass production and demonstrated that biotechnologically derived raw materials from *M. graeca* provide an efficient and innovative source of high-quality plant biomass rich in bioactive compounds, particularly phenolic acids. We further characterized the phenolic composition of methanolic extracts from these cultures using high-performance liquid chromatography (HPLC). The anticancer potential of extracts from 4-week-old cultures was assessed in HepG2 cells, comparing their efficacy to pure RA, as their most represented phenolic compound, and exploring mechanisms involving oxidative stress and apoptosis. Additionally, the antimicrobial activity of the extracts was tested against selected microbial strains. These findings contribute to the understanding of *M. graeca* as a source of bioactive compounds and its potential in pharmaceutical applications.

## 2. Results and Discussion

### 2.1. Micropropagation

#### 2.1.1. Shoot Culture Establishment

Proliferating shoot cultures of *M. graeca* were successfully established from surface-sterilized one-node stem segments (~1 cm) of wild-growing plants, with light green axillary buds emerging within 7–10 days after inoculation on basal MS medium [37].

#### 2.1.2. Shoot Multiplication

Both micropropagation via multiple axillary branching and adventive organogenesis offer prospects of faster multiplication. However, to reduce the risks of somaclonal variability during the multiplication of material, preformed apical and axillary meristems as sources of new plants are usually preferable to indirect organogenesis [38]. During regular subculturing, nodal explants grown on basal MS medium (AC−) consistently produced at least two axillary shoots within four weeks (Figure 1A), with occasional induction of secondary axillary buds on these shoots. These buds later developed into more or less elongated shoots (Figure 2A,B). However, the multiplication rate during a 4-week cultivation period (2.2 shoots per explant) was relatively low, compared to far more successful shoot multiplication (10–16 shoots per explant) in closely related species *Micromeria croatica* (Pers.) Schott [39] and *Micromeria pulegium* (Rochel) Benth. [40]. By contrast to *Micromeria* sp., a number of different Lamiaceae species required the addition of plant growth regulators (PGR) for shoot proliferation in vitro [41,42,43,44]. Prolonging the subculture interval to 5 weeks resulted in similar values of all measured parameters (Figure 1B and Figure 2). Number of shoots per explant, shoot length, and biomass production increased significantly only after a 6-week subculture interval (Figure 1C and Figure 2). Measured parameters did not change significantly even after 7 weeks of cultivation on AC− medium. Prolonged subculture intervals are known to reduce maintenance time and costs; however, a 6-week period was chosen for routine subculture because plantlets appeared healthier.

Relatively short adventitious roots formed spontaneously at the base of all explants already after 4 weeks of cultivation on basal medium without plant growth regulators (Figure 1A). A 6-week subculture period resulted in substantially more elongated and more branched adventitious roots, compared to shorter subculture intervals (Figure 1C).

Activated charcoal (AC), with its unique adsorption capacity, is often used in plant tissue culture to improve plant growth and development [26,29,30,45]. In shoot cultures of *Sideritis scardica* Griseb., the application of activated charcoal affected multiplication intensity similarly to PGR treatments [31]. In order to test the effect of activated charcoal on multiplication and biomass production in *M. graeca*, nodal segments were transferred to MS medium supplemented with 0.1% AC (AC+). During shorter subculture intervals (28 days), explants cultured on medium supplemented with 0.1% AC developed more axillary shoots compared to AC− plants (Figure 1D). However, in the presence of AC, callus often occurred at the explant base, while shoots were shorter and sometimes vitrified. Although prolonging the subculture interval to 6 weeks resulted in a significant increase in the number of shoots per explant in both media, the number of axillary shoots was considerably higher in AC− plants. The addition of AC to the culture medium did not stimulate shoot elongation in *M. graeca*, contrary to a number of species studied [26,46,47], or biomass production, regardless of the duration of the subculture interval. Adventitious roots that formed spontaneously on AC+ medium were considerably thicker, with significantly more lateral branches (Figure 1D).

The stimulatory effects of AC on morphogenesis are mainly due to its irreversible adsorption of inhibitory compounds in the culture medium, which substantially decreases the toxic metabolite levels, phenolic exudation, and brown exudate accumulation, but also due to the release of substances naturally present in AC that promote growth [45,48]. Although the majority of reports confirmed the positive role of AC in enhancing the growth and development of plant tissues [27,28,31,49,50], adverse effects were also reported in some systems [26,51]. In *M. graeca*, the addition of AC to the culture medium did not have beneficial effects, neither on multiplication nor biomass production during longer subculture intervals.

#### 2.1.3. Rooting and Acclimatization

Isolated axillary shoots spontaneously rooted on a PGR-free medium with a frequency of 100% after 6 weeks of in vitro cultivation. Rooted plantlets were successfully acclimatized (94% survival rate) upon transfer to ex vitro conditions. Acclimatized plants appeared healthy, showed no evidence of somaclonal variation, and did not exhibit morphological abnormalities (Figure 1E).

### 2.2. Determination of Phenolic Acids by HPLC

Specialized metabolite production often increases in response to environmental stressors or under specific developmental stages, diverting resources from primary growth. AC in the culture media likely adsorbs growth-promoting compounds (e.g., auxins or cytokinins) or phenolic inhibitors, shifting resource allocation from cell division and elongation to stress-related biosynthetic pathways. This trade-off, observed in tissue cultures, enhances specialized metabolites like phenolics or flavonoids, as AC mitigates oxidative stress or modifies medium chemistry [31,32]. To investigate these effects further, we analyzed the methanolic extracts of *M. graeca* shoots cultured on AC− or AC+ media for different periods of time.

The presence of phenolic acids in methanolic extracts of *M. graeca* shoots was confirmed by HPLC (Figure 3). In all chromatograms, the dominant peak at a retention time of 9.8 min was identified as rosmarinic acid, the most prominent phenolic compound characteristic of Lamiaceae species. Caffeic acid and syringic acid, corresponding to peaks 1 and 2, respectively, were also identified but at much lower concentrations. Based on the absorption maxima and shapes of the UV spectra, the three peaks eluting after rosmarinic acid correspond to derivatives of the identified phenolic acids. The peak at a retention time of 13.5 min is tentatively identified as a syringic acid derivative, while the other two peaks are tentatively identified as rosmarinic acid derivatives.

The quantification of the principal phenolic compounds found in methanolic extracts of *M. graeca* shoots revealed that AC+ shoots accumulated higher levels of phenolic acids than AC− shoots across subculture intervals (Table 1). Similarly, Danova et al. [31] demonstrated that AC not only promoted shoot growth of *S. scardica* but also influenced secondary metabolite production, specifically increasing the accumulation of phenylethanoids and flavone glycosides compared to wild-collected plants. In our system, caffeic and syringic acid contents were consistently elevated in AC+ shoots compared to AC− shoots. This difference was most pronounced for rosmarinic acid (RA), the predominant phenolic acid in *M. graeca* shoots. The highest level of RA was detected in AC+ shoots at a 4-week subculture interval, approximately threefold higher than in AC− shoots. Additionally, at a routine 6-week subculture interval, RA content in AC+ shoots remained substantially higher, exceeding fourfold that of AC− shoots.

Multiple studies have identified *Micromeria* species as a potent source of antioxidant polyphenols, with various phenolic acids, including gallic, chlorogenic, vanilic, hydroxybenzoic, rosmarinic, caffeic, syringic, coumaroylquinic, sinapic, ferulic, *p*-coumaric, cinnamic, and ellagic acids, detected in different species from natural populations [15,52,53,54,55,56,57,58,59]. A 2011 study by Vladimir-Knežević et al. [52] showed that dried samples of three selected species (*Micromeria thymifolia* (Scop.) Fritsch, *Micromeria juliana* (L.) Benth. ex Rchb., and *M. croatica*) contained considerable amounts of total polyphenols (9.69–13.06%), with the most abundant compounds being phenolic acids (5.26–6.84%). In a subsequent study of medicinal Lamiaceae species growing wild in Croatia, it was found that the main components of the tested extracts of *Micromeria* spp. were rosmarinic and chlorogenic acids, while caffeic and ferulic acids were generally less abundant [54]. The authors showed that rosmarinic acid, the predominant constituent of their ethanolic extracts, had a substantial influence on their acetylcholinesterase inhibitory and antioxidant activities. The bioactive methanolic extract of *Micromeria myrtifolia* Boiss. & Hohen., containing the highest levels of rosmarinic acid alongside quercetin and chlorogenic acid, displayed significant anticholinesterase and anti-urease activities [59].

Research on *M. graeca* extracts underscores its bioactive potential, particularly its phenolic constituents. Phenolic acids, including rosmarinic, chlorogenic, caffeic, ferulic, and gallic acids, were identified in *M. graeca* ethanolic extracts, with rosmarinic acid being the most abundant compound [15,54]. Several distinct studies specifically investigating *M. graeca* extracts covered various extract types (e.g., methanolic, ethanolic, aqueous, acetonic, essential oils) and their biological effects, such as antioxidant, antifungal, antibacterial, anti-hyperglycemic, anti-hypertensive, and anti-aflatoxigenic activities [12,13,15,17,60,61]. Unlike these studies, which focus on wild or greenhouse plants, our study of *M. graeca* in vitro shoot cultures revealed increased rosmarinic acid content in methanolic extracts from AC-supplemented media, suggesting a novel approach to optimizing bioactive yields. This underscores the pharmaceutical potential of in vitro grown *M. graeca*, particularly for polar extracts.

### 2.3. Antimicrobial Activity

The antimicrobial activities of different *M. graeca* methanolic extracts assayed against microbial strains tested in the present study were quantitatively assessed by the minimum inhibitory concentration (MIC) values. The results showed that the methanolic extracts from the aerial parts of in vitro cultured *M. graeca* did not exhibit any significant antimicrobial activity compared to reference antibiotics and fungicide (Table 2).

Ethanolic extract of *M. graeca* also exhibited a low antibacterial effect, although the combination of this extract with antibiotics restored the activities of cefotaxime and streptomycin against resistant *Staphylococcus aureus* and *Pseudomonas aeruginosa* [15]. This suggests that *M. graeca* extract supplementation could serve as an adjuvant therapy for infections caused by resistant bacterial strains.

Plant extracts generally exhibit significantly lower antimicrobial activity than essential oils, which was confirmed by multiple studies [62,63,64]. A comparative study of the antimicrobial activity of *Micromeria fruticosa* (L.) Druce essential oil and methanolic extract against 40 microbial strains demonstrated that the essential oil exhibited markedly superior antimicrobial activity against yeast and fungal species, whereas the methanolic extract showed no activity against any strain tested [63]. Similarly, *M. graeca* essential oil exhibited varying degrees of antibacterial activities against all tested strains and a pronounced anticandidal effect against *Candida albicans* [16]. However, *M. graeca* extracts obtained using different solvents did not significantly affect the growth of *C. albicans* and *Aspergillus niger* [17,61], whereas the aqueous extract of *M. graeca* completely inhibited aflatoxin B1 (AFB1) production without affecting fungal growth [60].

The antimicrobial activity of *Micromeria* species has primarily been studied in wild-growing plants [15,17,61,63,65,66,67]. For wild-growing *Micromeria nervosa* (Desf.) Benth., different extracts varied significantly in their antimicrobial activities, with ethanolic extracts showing the highest activity, followed by ethyl acetate and petroleum ether extracts [65]. For *M. pulegium*, both wild-growing and in vitro-cultured plant extracts showed comparable antibacterial activity, though significantly less potent than reference antibiotics [68]. In contrast, *M. croatica* methanolic extracts from in vitro cultures demonstrated higher antibacterial activity, especially against Gram-positive bacteria, compared to wild-growing plants [39]. However, methanolic extracts from in vitro cultured *M. graeca*, grown under the present study conditions, showed negligible antimicrobial activity. This suggests that while optimized in vitro culture conditions can enhance antimicrobial compound production in certain *Micromeria* species, the efficacy of extracts depends on both species and extraction method, highlighting the selective potential of in vitro shoot cultures as a source of antimicrobial agents.

### 2.4. Anti-Hepatoma Activity

Rosmarinic acid (RA), the predominant phenolic acid in *M. graeca* shoots, has been demonstrated as a promising candidate for anticancer activity [33,35]. The highest RA levels were detected in the methanolic extract from *M. graeca* shoots cultured on AC+ medium for four weeks (Mg+). Consequently, Mg+ extract was selected for HepG2 cytotoxicity testing alongside extract from plants cultured on AC− medium (Mg−) and pure rosmarinic acid to evaluate their efficacy.

#### 2.4.1. Mg Extracts and RA Exert Anti-Hepatoma Effect Against HepG2 Cells

To investigate the anti-hepatoma potential of RA-rich *M. graeca* extracts (Mg), human HepG2 hepatocellular carcinoma cells were incubated with different concentrations of Mg− and Mg+ extracts (100–400 µg mL^−1^) for 48 h. MTT and CV tests have shown that Mg− extract was more potent in the reduction of mitochondrial activity and the number of adherent, viable HepG2 cells than Mg+ extract (Figure 4A,B). IC50 value determined for Mg− extract was 193 µg mL^−1^, while Mg+ extract displayed an IC50 value of 274.7 µg mL^−1^ (CV). Considering the high content of RA detected in Mg extracts, the anti-hepatoma activity of pure RA was next examined. For this purpose, HepG2 cells were treated with RA at concentrations ranging from 25 to 200 µM for 48 h. RA slightly inhibited mitochondrial activity and lowered the number of adherent HepG2 cells, as shown by MTT and CV assays (IC50(RA) = 380.7 µM; CV) (Figure 4C). However, even at the highest applied concentration, RA did not reach the level of viability reduction detected with the Mg− or Mg+ extracts. Specifically, at the highest applied RA dose (200 µM), cell viability was reduced to approximately 80%. In contrast, treatment with 400 µg mL^−1^ Mg+ extract (containing 180 µM RA, per Table 1) reduced viability to ~40%, while 400 µg mL^−1^ Mg− extract (containing 60 µM RA) reduced viability to below 20%. These findings suggest that the cytotoxic effects of the extracts are not solely dependent on RA concentration, indicating that other extract components or synergistic interactions likely contribute to the observed effects.

#### 2.4.2. Mg Extracts Induce Oxidative Stress and Apoptosis, RA Inhibits the Proliferation of HepG2 Cells

To further determine the type and mechanism of HepG2 cell death, cells were exposed to Mg− and Mg+ extracts at different concentrations (100–400 µg mL^−1^) and the highest investigated RA concentration (200 µM) for 48 h. Flow cytometry analysis of Annexin-V/PI stained cells revealed that both Mg extracts elevated the proportion of apoptotic Ann+PI- cells in a dose-dependent manner, with a minor shift toward necrotic Ann+PI+ cells (Figure 5A). In line with the results of cell viability tests, the Mg− extract exhibited a more pronounced pro-apoptotic effect compared to the Mg+ extract, whereas RA had no impact on the live/dead cell ratio (Figure 5A). Given that apoptosis induction is associated with oxidative stress [69], we subsequently investigated the effects of Mg extracts and RA on intracellular reactive oxygen species (ROS) levels. The Mg+ extract, and particularly the Mg− extract, induced a dose-dependent increase in total intracellular ROS levels, as demonstrated by flow cytometric analysis of DHR-stained cells after 24-h treatment (Figure 5B). In contrast, RA did not alter the proportion of DHR-positive cells (Figure 5B). To elucidate the mechanism underlying RA-induced reduction in cell viability, cells were incubated with RA (25–200 µM) for 4 days, and viability was assessed using CV and MTT assays. RA effectively suppressed the growth of HepG2 cells, indicating anti-proliferative rather than cytotoxic effects (IC50 = 51.62 µM; CV assay, 96 h; Figure 5C). These findings suggest that Mg extracts induce HepG2 cell apoptosis associated with oxidative stress, whereas RA inhibits HepG2 cell proliferation.

In this study, we identified modest anti-hepatoma activity of methanolic extracts derived from in vitro cultured *M. graeca* shoots. This finding aligns with studies reporting moderate or negligible activity of alcoholic/polar extracts from different *Micromeria* species against various cancer cell lines, including HepG2 cells [36,70]. In contrast, the more potent cytotoxic activity of *Micromeria* extracts reported in other studies [71,72] may result from the use of different *Micromeria* species or a cell-type-specific response associated with distinct cancer cell lines. Our results showed that at high doses, both Mg extracts induced oxidative stress and apoptosis in HepG2 hepatoma cells, likely attributable to their elevated phenolic compound content. The observed activity of investigated extracts is consistent with studies showing a pro-oxidant effect of plant polyphenols at high concentrations [73], which may induce DNA damage or mitochondrial toxicity, resulting in cancer cell apoptosis [74]. A comparable cytotoxic effect, driven by the pro-oxidant activity of polyphenol-rich *M. croatica* methanolic extract, was previously observed in human laryngeal carcinoma cells [75]. Notably, in our study, the Mg− extract exhibited stronger anti-hepatoma activity, including enhanced induction of oxidative stress and apoptosis in HepG2 cells, than the Mg+ extract, despite its lower total phenolic content. Therefore, the enhanced cytotoxicity effect of the Mg− extract compared to Mg+ extract and particularly pure RA, may not stem solely from its major constituent but likely involves compounds beyond rosmarinic acid. It is well-established that the efficacy of compounds in a mixture can differ from that of their isolated components and that the biological activity of the mixture may not be solely attributable to its primary compound, as synergistic interactions among minor compounds may also play a role [16,76,77]. Another potential contributor could be monoterpenes, secondary metabolites prevalent in *Micromeria* species [78,79,80,81], which have been shown to promote oxidative stress and apoptosis in cancer cell lines [82].

Our study demonstrated qualitatively different effects of RA-rich Mg extracts and RA on hepatoma cells, with RA inhibiting HepG2 cell proliferation rather than inducing apoptosis. Although previous studies have reported significant anticancer properties of RA, including pro-apoptotic and anti-proliferative effects against SMMC-7721 hepatoma, breast, and melanoma cancer cell lines [83,84,85,86], we observed no evidence of early or late apoptosis following RA treatment in our experimental setting. This discrepancy may result from RA’s differential regulation of apoptosis-related gene expression across various cancer cell lines, leading to divergent cell death outcomes [84]. Taken together, our findings suggest that the modest anti-hepatoma activity of *M. graeca* methanolic extracts is independent of RA.

## 3. Materials and Methods

### 3.1. Micropropagation of Micromeria graeca

#### 3.1.1. Plant Material, Shoot Culture Establishment and Maintenance

Shoot cultures of *M. graeca* were established according to the procedure described by Uzelac et al. [37]. Namely, aerial parts of *M. graeca* were randomly collected from the natural population in Longos (Paxos, Greece), at latitude 39°13′41.93″ N and longitude 20°9′40.37″ E in July 2011. Voucher specimen (No. 14623) was deposited in the Herbarium collection of the Faculty of Sciences and Mathematics, University of Niš (HMN).

Briefly, shoots of wild-growing plants were dissected into one-node stem segments (~1 cm) bearing two axillary buds and used as explants to establish in vitro cultures. Nodal segments were surface-sterilized with 25% (*v*/*v*) commercial bleach in sterile deionized water (1.5% (*w*/*v*) active chlorine) containing two drops of liquid commercial detergent for 30 min and rinsed three times with sterile deionized water. Explants were then placed in the solution containing 500 mg L^−1^ Nystatin for 24 h, followed by three rinses with sterile deionized water. Ten explants were placed in each 370-mL glass jar containing 50 mL of basal Murashige and Skoog (MS) medium [87] supplemented with 3% (*w*/*v*) sucrose and 0.7% (*w*/*v*) agar (Torlak, Belgrade). The pH of the media was adjusted to 5.8 with 1 N NaOH prior to autoclaving at 114 °C for 25 min. Cultures were maintained in a growth chamber under a 16-h photoperiod, with a photon flux density of 45 µmol m^−2^ s^−1^ provided by cool white, fluorescent tubes, at 25 ± 2 °C.

After four weeks of cultivation, at least two axillary shoots developed from each surviving explant. Upon establishing aseptic culture, uniform nodal segments of 4-week-old plantlets were used as explants for the multiplication, in order to obtain enough plant material for experiments. The stock shoot cultures were maintained on basal MS medium, under previously described growth conditions, and subcultured on fresh medium every 6 weeks.

#### 3.1.2. Shoot Multiplication and Biomass Production

Nodal segments derived from shoot cultures grown on basal medium for six weeks were transferred to MS medium supplemented with 0.1% activated charcoal (AC+) or without it (AC−), and cultured for the following different periods of time: 4, 5, 6, or 7 weeks. Ten explants were placed in each 370-mL glass jar containing 50 mL of the respective medium and maintained under previously described growth conditions.

The effect of activated charcoal and varying subculture intervals on growth was determined by assessing the number of explants producing shoots and the number of shoots per explant, and by recording the shoot length as well as explant fresh and dry weight following each treatment. For weight measurements, the aerial parts of the plants were used. Fresh weight was recorded upon removal from the culture (on the day of sampling), whereas dry weight was recorded after drying the explants at 60 °C for 72 h.

#### 3.1.3. Acclimatization

Explants cultured for 6 weeks on basal MS that spontaneously rooted were removed from the medium. Roots and basal stem parts were washed with water to remove any adherent culture medium. Plantlets were transferred to plastic pots with substrate (“Substral”, Serbia). Pots were enclosed with a thin transparent plastic envelope to reduce water loss and maintained in the greenhouse with high humidity (70–80%). Plantlets were gradually acclimatized over 4 weeks by daily removal of the thin envelope in increasing intervals until final acclimatization. The survival of the plants was evaluated after 8 weeks.

### 3.2. HPLC Analysis

#### 3.2.1. Preparation of Methanolic Extracts

Methanolic extracts of *M. graeca* were obtained from the aerial parts of in vitro plants cultured on AC+ or AC− media for varying durations (4, 5, 6, or 7 weeks).

The air-dried plant material (about 10 g) was ground to a fine powder and extracted with 100 mL of methanol (HPLC grade, Merck, Darmstadt, Germany) by ultrasonic extraction for 20 min. The extraction was continued by maceration for 48 h in the dark at room temperature. After filtration, the residue was extracted again with 100 mL of methanol and macerated for 48 h at room temperature in the dark. The extracts were combined and concentrated to dryness at 50 °C in a vacuum rotary evaporator (Buchi R-210, Flawil, Switzerland). The dried methanolic extracts were stored at 4 °C until use.

#### 3.2.2. HPLC Conditions

Prior to HPLC analysis, the dried methanolic extracts (10 mg) were dissolved in 1 mL of methanol and filtered through nylon syringe filters (Captiva, 0.45 μm pore size, 13 mm; Agilent Technologies, Waldbronn, Germany).

The identification and quantification of phenolic compounds was performed on an Agilent 1100 series HPLC instrument (Agilent, Waldbronn, Germany) with a DAD detector on a Zorbax SB-C18 (Agilent Technologies, Waldbronn, Germany) reversed-phase analytical column (250 mm × 4.6 mm i.d., 5 μm particle size). The mobile phase consisted of solvent A (0.1%, *v*/*v* solution of ortho-phosphoric acid in water) and solvent B (acetonitrile), and the elution gradient was as follows: 75–73% A 0–10 min, 73–65% A 10–20 min, 65–30% A 20–25 min, 30–0% A 25–32 min. The temperature was maintained at 25 °C. The wavelengths for detection were set to 280 and 330 nm, and the flow rate was set to 0.9 mL min^−1^. The injection volume of the samples was 10 μL.

Rosmarinic acid (RA), caffeic acid (CAFFA), and syringic acid (SYRA) were identified using standard compounds purchased from Sigma–Aldrich (Steinheim, Germany). The quantification was performed using calibration curves with external standard compounds. The calibration curves for rosmarinic and caffeic acids were recorded at 330 nm, and for syringic acid at 280 nm. The stock solutions of phenolic acid standards were prepared by dissolving each compound in methanol. Linear regression analyses of calibration curves of standards showed linearity with a correlation coefficient of r^2^ = 0.9986–0.9999. The amounts of the analyzed phenolic acids were expressed as mg per g dry weight of extract (mg g^−1^ DW).

### 3.3. Antimicrobial Activity

#### 3.3.1. Microbial Strains

In vitro antimicrobial activity was tested against the following bacterial pathogenic strains from ATCC (American Type Culture Collection): Gram (−) bacteria—*Escherichia coli* ATCC 25922, *Pseudomonas aeruginosa* ATCC 9027, *Salmonella enteritidis* ATCC 13076, and *Enterobacter aerogenes* ATCC 13048; Gram (+) bacteria—*Staphylococcus aureus* ATCC 25923, *Bacillus cereus* ATCC 11778, and *Enterococcus faecalis* ATCC 29212; and yeast *Candida albicans* ATCC 10231, using Micro-well Dilution Assay.

#### 3.3.2. Micro-Well Dilution Assay

The determination of the minimum inhibitory concentration (MIC) and minimum microbicidal concentration (MMC) of samples was carried out using Micro-well Dilution Assay, according to the Clinical and Laboratory Standards Institute (CLSI [88]) with some modifications. Overnight cultures of the tested strains were each used to prepare a 0.5 McFarland standard turbidity suspension (corresponding to 10^8^ CFU/mL). All tested samples were dissolved in dimethyl sulfoxide (DMSO). Serial doubling dilutions of tested samples (Table 3) were prepared in 96-well microtiter plates with inoculated Mueller–Hinton broth (MHB) for bacteria and Sabouraud Dextrose broth (SDB) for yeast.

The final volume was 100 µL, and the final concentration of bacterial cells was 10^6^ CFU/mL in each well. The plates were incubated for 24 h at 37 °C. DMSO solvent at concentrations ranging from 100 to 0.05% (*v*/*v*) was used as a negative control. All determinations were performed in triplicate. Microbial growth was determined by adding 20 μL of 0.5% triphenyl tetrazolium chloride (TTC) aqueous solution. MIC was defined as the lowest concentration of the tested samples at which microbial strains showed no visible growth. In order to determine MMC, broth was taken from each well and inoculated on Mueller–Hinton agar (MHA) for bacteria and Sabouraud Dextrose agar (SDA) for yeast. The plates were incubated for 24 h at 37 °C. The MMC is defined as the lowest concentration of the tested samples at which 99.9% of inoculated microorganisms were killed.

### 3.4. Cytotoxic Activity of M. graeca Methanolic Extracts Against HepG2 Cell Line

#### 3.4.1. Cell Line

The human hepatocellular carcinoma HepG2 cell line was obtained from the European Collection of Authenticated Cell Cultures (ECACC, UK Health Security Agency, Salisbury, UK; Catalogue No. 85011430). Cells were cultivated in a high glucose Dulbecco’s Modification of Eagle’s Medium supplemented with 10% fetal bovine serum (FBS), 2 mM L-glutamine, 1 mM sodium pyruvate, 100 U mL^−1^ penicillin, and 100 μg mL^−1^ streptomycin (all purchased from Capricorn scientific, Ebsdorfergrund, Germany). They were maintained in a humidified atmosphere with 5% CO_2_ at 37 °C and prepared for the experiments using a conventional trypsinization procedure. The seeding density was 1000 or 5000 cells/well in 96-well microplates for cell viability evaluation, and 40,000 cells/well in 24-well microplates for flow cytometric analyses.

#### 3.4.2. Treatments

*Micromeria graeca* methanolic extracts (Mg) (50 mg mL^−1^) and rosmarinic acid (RA) (50 mM) were stored as stock solutions in DMSO at +4 °C and diluted in culture medium on the day of use. The cells were treated as described in the Results and Figure legends.

#### 3.4.3. Cell Viability Assays

In order to obtain a more accurate assessment of cell viability, two assays based on the intrinsic characteristics of adherent viable cells, cell adherence (crystal violet (CV) test) and mitochondrial activity (3-(4,5-dimethylthiazol-2-yl)-2,5-diphenyltetrazolium bromide (MTT) reduction assay), were performed as previously described [89]. The staining of attached cells with CV dye, which binds to proteins and DNA, is used to detect maintained adherence of cells in a culture. Since cells that undergo cell death lose their adherence, they are subsequently lost from the population of cells, thus reducing the amount of crystal violet staining in a culture. Therefore, the amount of CV staining correlates with the number of living cells in the plates. On the other hand, only viable and metabolically active cells with active mitochondria can metabolize MTT to its purple-colored formazan salt.

Briefly, cells that remained attached after treatment were fixed with methanol and incubated in CV solution (0.1%) at room temperature for 15 min. The crystal violet dye bound to the adherent cells was dissolved in acetic acid (33%), and the optical density was measured using a microplate reader at 570 nm (Multiskan, Thermo Fisher Scientific, Waltham, MA, USA). Alternatively, MTT solution (0.5 mg mL^−1^) was incubated with the cells at 37 °C for 1 h after treatment. The purple formazan salts formed in the cells with active mitochondrial dehydrogenases were dissolved in DMSO, and the optical density was determined at 570 nm (Multiskan, Thermo Fisher Scientific, Waltham, MA, USA). The measured absorbance corresponded to the number of viable, adherent cells (CV) or the number of cells with active mitochondria (MTT). The results are expressed as a percentage of untreated control cells considered as 100% viable (Figure 4), or as a fold increase compared to the cells seeded at the start of the experiment (Figure 5).

#### 3.4.4. Flow Cytometric Analyses

##### Apoptosis/Necrosis Assessment

Apoptosis and necrosis were analyzed after staining cells with fluorescein isothiocyanate (FITC)-conjugated Annexin-V (Ann) and propidium iodide (PI) dyes, according to the manufacturer’s instructions (Elabscience, Houston, TX, USA). Annexin-V binds to phosphatidylserine exposed on cells in the initial apoptotic phase, whereas PI labels only cells with damaged membranes in the late apoptotic/necrotic phase. Cells were detached from wells and incubated with Ann (2 μg mL^−1^) and PI (20 μg mL^−1^) for 15 min at 37 °C in the dark. Green (FL1) and red (FL2) fluorescence was measured using Partec CyFlow© Space flow cytometer with FloMax software version 2.3 (Partec, Canterbury, UK) to determine the number of live (Ann^−^/PI^−^), early apoptotic (Ann^+^/PI^−^), and late apoptotic/necrotic (Ann^+^/PI^+^) cells.

#### 3.4.5. Intracellular ROS Production

The intracellular levels of total ROS were determined after staining cells with the redox-sensitive dye dihydrorhodamine 123 (DHR). DHR freely penetrates the cell, where it is oxidized by ROS to its green, fluorescent derivative. The cells were incubated with DHR (5 μM) during treatment and detached from the surface after incubation. The mean intensity of green fluorescence (FL1), corresponding to the total ROS levels, was measured by flow cytometry (Partec, Canterbury, UK).

### 3.5. Statistical Analysis

For micropropagation experiments, three replicates, each consisting of ten explants, were used per treatment, and the experiment was repeated (*n* = 60). The data were averaged and statistically analyzed, and differences were tested for significance using one-way ANOVA and Tukey’s HSD test at the significance level of *p* ≤ 0.05.

The statistical analysis of the results of bioactivity assays was performed using one-way ANOVA followed by the Student–Newman–Keuls test at the significance level of *p* ≤ 0.05.

## 4. Conclusions

This study demonstrates that in vitro culture conditions, particularly the use of activated charcoal, enhance phenolic acid accumulation in *Micromeria graeca* shoot cultures, with rosmarinic acid as the dominant compound. While antimicrobial activity was negligible, extracts from 4-week-old cultures showed modest anti-hepatoma activity via oxidative stress-induced apoptosis, independent of rosmarinic acid. These findings highlight the potential of optimized in vitro systems to enhance bioactive compound production for pharmaceutical applications, with further optimization of culture conditions and exploration of synergistic compound interactions potentially improving the therapeutic utility of *M. graeca* extracts.

## Figures and Tables

**Figure 1 plants-14-02592-f001:**
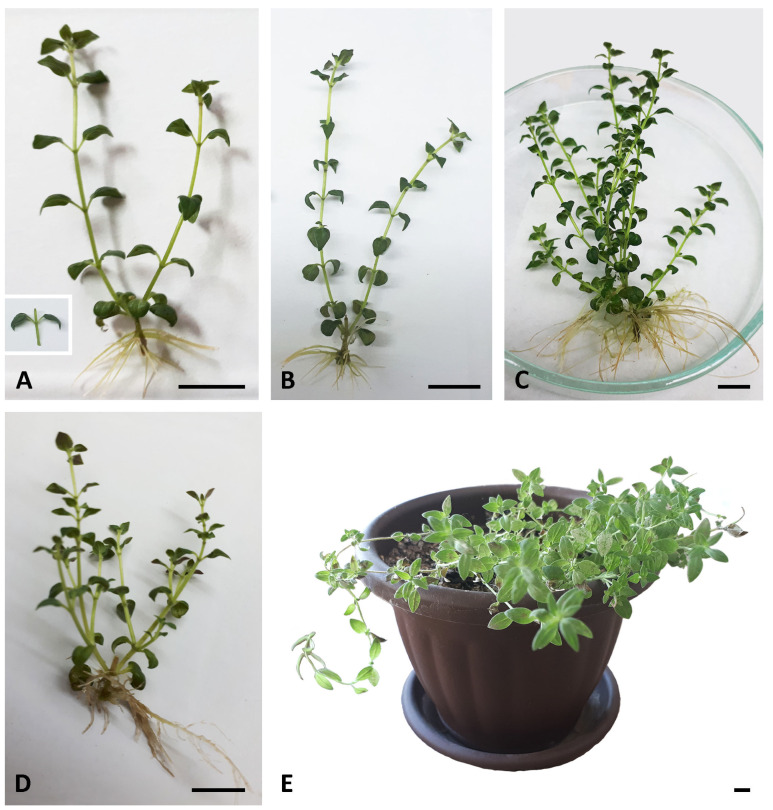
Micropropagation of *Micromeria graeca*. (**A**) Initial nodal explant (left) and non-branched shoots developed in its leaf axils (right) after 4 weeks of culture on basal MS medium. Shoots bear ovate leaves, whose sharply pointed tips are bent downwards. (**B**,**C**) In vitro plantlets of *M. graeca* obtained after 5 (**B**) and 6 (**C**) weeks of culture on basal medium. Note numerous axillary shoots and well-developed adventitious roots in the 6-week-old shoot culture in (**C**). (**D**) A 4-week-old plantlet cultured on medium supplemented with 0.1% activated charcoal, with numerous thick adventitious roots. (**E**) The acclimatized plant regenerated via axillary shoots after 9.5 weeks of acclimatization. Scale bars: 1 cm.

**Figure 2 plants-14-02592-f002:**
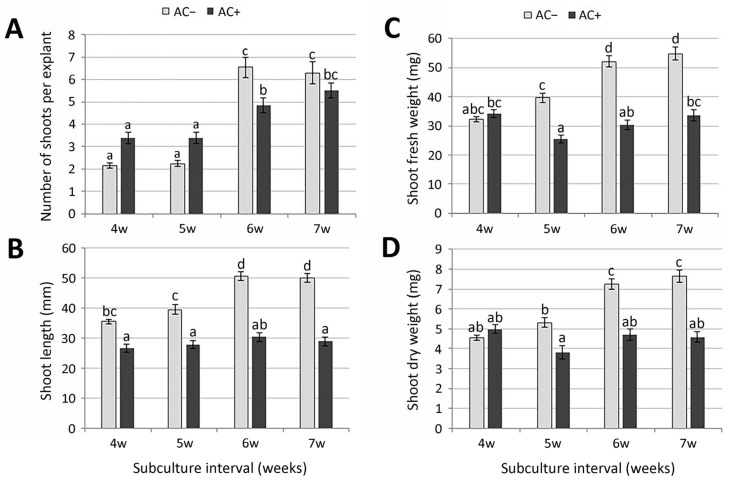
Effect of subculture interval and activated charcoal (AC) on multiplication and biomass production of in vitro propagated *Micromeria graeca*: number of shoots per explant (**A**), shoot length (**B**), and shoot fresh (**C**) and dry (**D**) weight. Values represent means ± SE, *n* = 60. Means within the same histogram marked with different letters are significantly different according to Tukey’s HSD test at *p* ≤ 0.05.

**Figure 3 plants-14-02592-f003:**
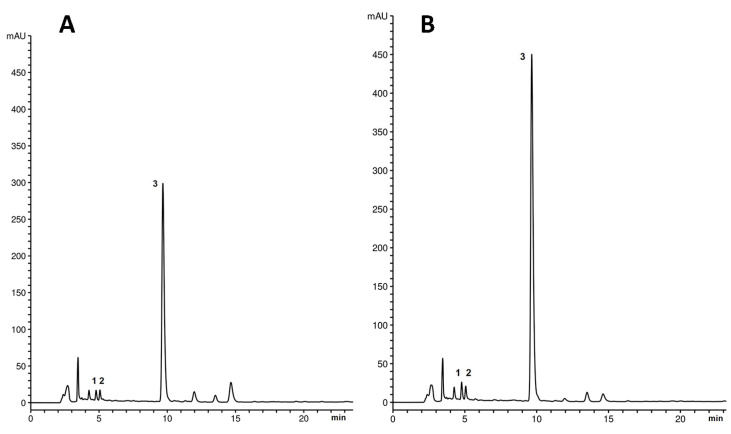
Comparative chromatograms of methanolic extracts from *Micromeria graeca* shoots cultured in vitro for 4 weeks on MS medium without (**A**) or with 0.1% activated charcoal (**B**), recorded at 280 nm. Phenolic acids identified correspond to caffeic acid (1), syringic acid (2), and rosmarinic acid (3).

**Figure 4 plants-14-02592-f004:**
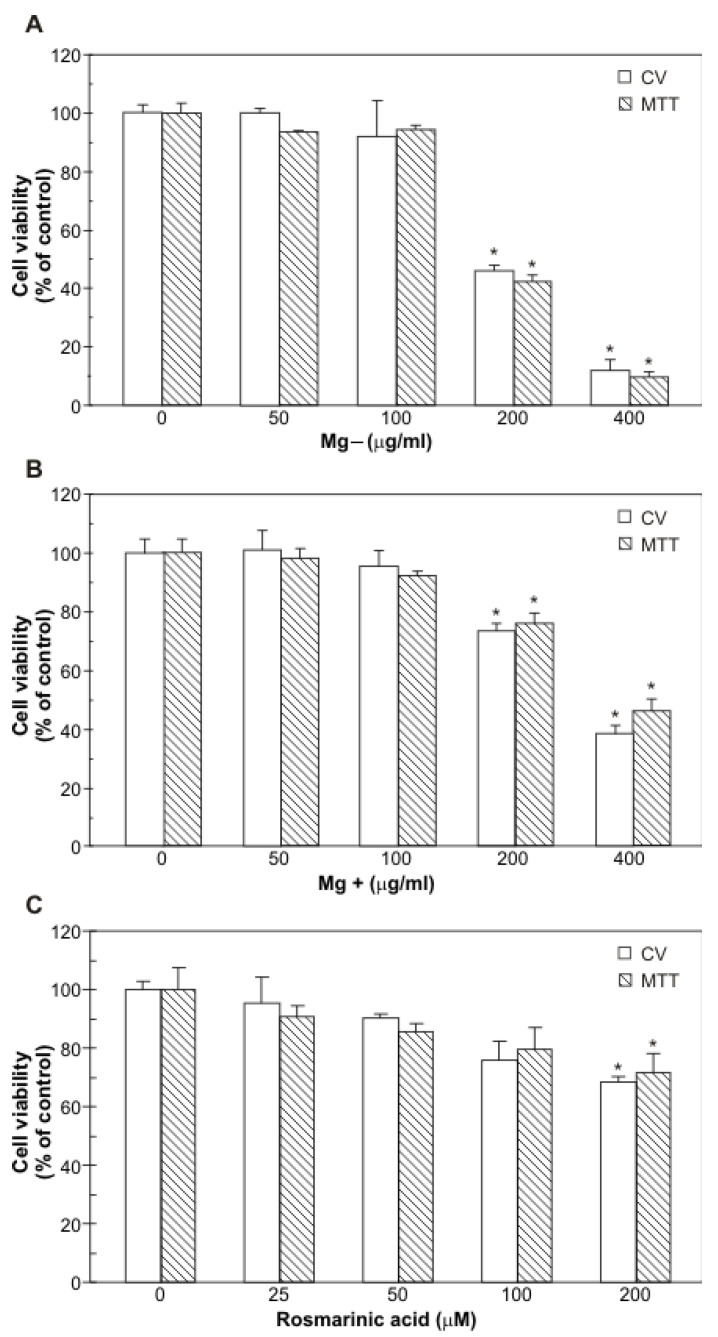
The effect of Mg extracts and rosmarinic acid on the viability of HepG2 cells. Cell viability was assessed using CV and MTT assays, after 48-h incubation of HepG2 cells with different concentrations of Mg− (**A**) or Mg+ (**B**) extracts, or rosmarinic acid (**C**). Data are presented as means ± SD of triplicates from three independent experiments. Asterisks denote statistically significant differences compared to untreated, control cells (*p* ≤ 0.05).

**Figure 5 plants-14-02592-f005:**
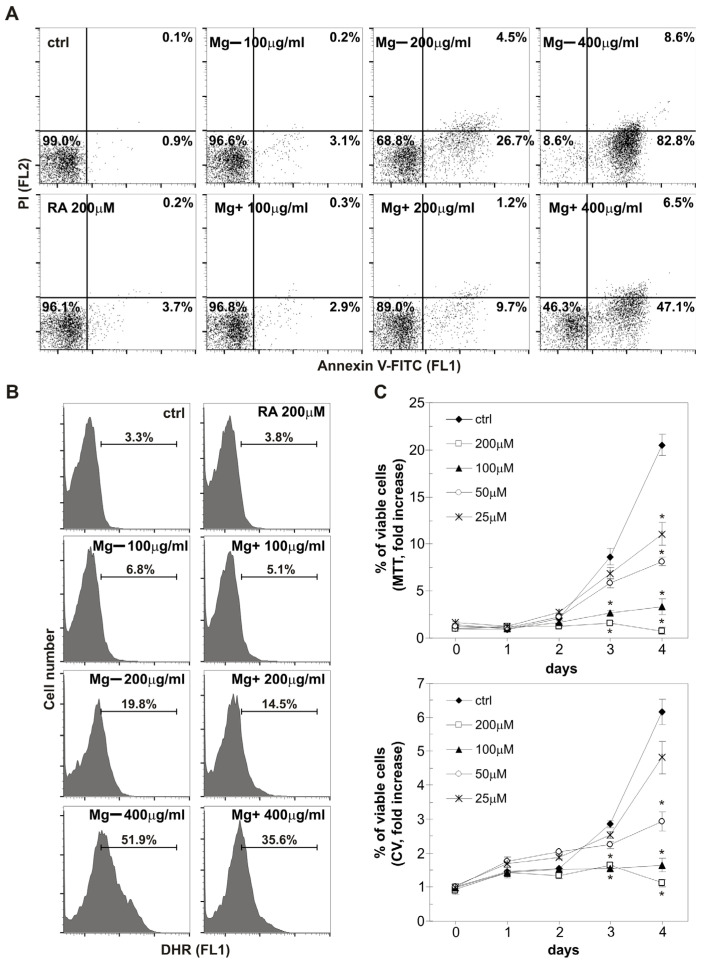
Effects of Mg extracts and rosmarinic acid on HepG2 cell apoptosis, oxidative stress, and proliferation. HepG2 cells were treated with Mg− or Mg+ extracts (100 to 400 µg mL^−1^) or rosmarinic acid (200 µM) for 48 h (**A**) or 24 h (**B**), or with rosmarinic acid (25 to 200 µM) for 4 days (**C**). Apoptosis and necrosis were assessed using Annexin-V/PI staining (**A**), and total intracellular ROS levels were measured using DHR staining (**B**), both by flow cytometry. Cell viability was evaluated using CV and MTT assays (**C**). Data are presented as representative dot plots (**A**), histograms (**B**), or means ± SD of triplicates (**C**) from one of three independent experiments. Asterisk denotes statistically significant (*p* ≤ 0.05) difference compared to untreated, control cells.

**Table 1 plants-14-02592-t001:** Effect of activated charcoal and treatment duration on phenolic acid content in methanolic extracts obtained from *Micromeria graeca* shoots grown in vitro.

Shoot Culture Designation *	Subculture Interval (Weeks)	Caffeic Acid (mg/g DW Extract)	Syringic Acid (mg/g DW Extract)	Rosmarinic Acid (mg/g DW Extract)
AC−	4	1.52 ± 0.05 ^a^	0.52 ± 0.05 ^a^	51.68 ± 0.26 ^b^
	5	1.68 ± 0.07 ^ab^	0.54 ± 0.02 ^a^	51.39 ± 0.15 ^b^
	6	1.34 ± 0.07 ^a^	0.55 ± 0.03 ^a^	28.92 ± 0.54 ^a^
	7	1.69 ± 0.03 ^ab^	0.81 ± 0.02 ^b^	52.77 ± 0.65 ^b^
AC+	4	2.17 ± 0.07 ^c^	0.78 ± 0.07 ^b^	157.75 ± 0.79 ^d^
	5	3.54 ± 0.08 ^e^	1.37 ± 0.04 ^d^	123.24 ± 0.44 ^c^
	6	2.93 ± 0.09 ^d^	1.15 ± 0.01 ^c^	122.21 ± 0.47 ^c^
	7	2.02 ± 0.05 ^bc^	0.93 ± 0.01 ^b^	52.40 ± 0.23 ^b^

* Shoots cultured on basal MS medium supplemented with activated charcoal at 0 g L^−1^ (AC−) or 1 g L^−1^ (AC+). Values represent means ± SE, *n* = 3. Means within the same column marked with different letters are significantly different according to Tukey’s HSD test at *p* ≤ 0.05.

**Table 2 plants-14-02592-t002:** Minimum inhibitory concentrations (MICs) of methanolic extracts (mg mL^−1^) obtained from in vitro cultured *Micromeria graeca* against selected microbial strains, determined by the broth microdilution method.

Microbial Strains from ATCC Collection	MICs (mg mL^−1^) of *M. graeca* Methanolic Extracts								MIC/MMC * (µg mL^−1^) of Reference Antibiotics ^1^		MIC/MMC (µg mL^−1^) of Reference Fungicide ^2^
	AC+				AC−				*Str*	*Chl*	*Nys*
	4w	5w	6w	7w	4w	5w	6w	7w			
Gram (−) bacteria											
*Escherichia coli* ATCC 25922	>54.68	>44.10	>59.20	>75.20	>68.80	>43.90	>57.50	>58.20	16.0/16.0	16.0/16.0	-
*Pseudomonas aeruginosa* ATCC 9027	>54.68	>44.10	>59.20	>75.20	>68.80	>43.90	>57.50	>58.20	8.0/8.0	4.0/16.0	-
*Salmonella enteritidis* ATCC 13076	>54.68	>44.10	>59.20	>75.20	>68.80	>43.90	>57.50	>58.20	4.0/4.0	4.0/8.0	-
*Enterobacter aerogenes* ATCC 13048	>54.68	>44.10	>59.20	>75.20	>68.80	>43.90	>57.50	>58.20	0.5/0.5	1.0/1.0	-
Gram (+) bacteria											
*Staphylococcus aureus* ATCC 25923	>54.68	>44.10	>59.20	>75.20	>68.80	>43.90	>57.50	>58.20	0.5/0.5	1.0/8.0	-
*Bacillus cereus* ATCC 11778	>54.68	>44.10	>59.20	>75.20	>68.80	>43.90	>57.50	>58.20	0.5/0.5	1.0/4.0	-
*Enterococcus faecalis* ATCC 29212	>54.68	>44.10	>59.20	>75.20	>68.80	>43.90	>57.50	>58.20	16.0/16.0	16.0/16.0	-
Yeast											
*Candida albicans* ATCC 10231	>54.68	>44.10	>59.20	>75.20	>68.80	>43.90	>57.50	>58.20	-	-	16.0/16.0

* MMC—minimum microbicidal concentration; ^1^—*Str* Streptomycin, *Chl* Chloramphenicol; ^2^—*Nys* Nystatin.

**Table 3 plants-14-02592-t003:** Serial doubling dilutions of tested methanolic extracts obtained from in vitro cultured *Micromeria graeca* used for the micro-well dilution assay.

Shoot Culture Designation *	Subculture Interval (Weeks)	Sample Dilution Range (mg mL^−1^)
AC+	4	54.68–0.03
	5	44.10–0.02
	6	59.20–0.03
	7	75.20–0.04
AC−	4	68.80–0.03
	5	43.90–0.02
	6	57.50–0.03
	7	58.20–0.03

* Shoots cultured on basal MS medium supplemented with activated charcoal at 0 g L^−1^ (AC−) or 1 g L^−1^ (AC+).

## Data Availability

Data will be made available upon request to interested researchers.

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
