# Peer review of "In Vitro Shoot Cultures of Micromeria graeca: Micropropagation and Evaluation of Methanolic Extracts for Anticancer and Antimicrobial Activity"

_plants, 2025, doi:10.3390/plants14162592_

Round 1

Reviewer 1 Report

Comments and Suggestions for Authors

The revised manuscript “In Vitro Shoot Cultures of Micromeria graeca: Micropropagation and Evaluation of Methanolic Extracts for Anticancer and Antimicrobial Activity”, by Branka Uzelac et al., is an original, clear, ordered and well-presented research study.

The manuscript describes the micropropagation process of Micromeria graeca under several conditions including variables as charcoal activated presence and times of propagation. Then the phytochemical profile was obtained and compared between the propagation conditions. In addition, biological properties such as antimicrobial, cytotoxic and antioxidant activities were studied and compared. The results obtained support the conclusion and provide a valuable precedent in the study of in vitro systems to enhance bioactive compounds production related to plants with ethnopharmacological reported uses.

Some minor modifications are listed below:

  • Line 63. Please include in the introduction the most significant/representative anti-enzymatic effects.
  • Line 224: “P ≤ 0.05”, “p” must appear in lower case letter. Same in line 332 and others.
  • Line 228: “p-coumaric”, “p” must appear in italics. In addition, all the phytochemicals must be initialized with capital letters, i.e. Gallic acid. Change the use of capital letters in phytochemicals along the text.

In conclusion, in accordance to this reviewer criteria the manuscript in able to be published in the Plants Journal after minor modifications.

Author Response

Response to Reviewer 1 Comments

Comment 1:

The revised manuscript “In Vitro Shoot Cultures of Micromeria graeca: Micropropagation and Evaluation of Methanolic Extracts for Anticancer and Antimicrobial Activity”, by Branka Uzelac et al., is an original, clear, ordered and well-presented research study.

The manuscript describes the micropropagation process of Micromeria graeca under several conditions including variables as charcoal activated presence and times of propagation. Then the phytochemical profile was obtained and compared between the propagation conditions. In addition, biological properties such as antimicrobial, cytotoxic and antioxidant activities were studied and compared. The results obtained support the conclusion and provide a valuable precedent in the study of in vitro systems to enhance bioactive compounds production related to plants with ethnopharmacological reported uses.

Response 1: We sincerely thank you for your time and valuable feedback in reviewing our manuscript. Detailed responses to your comments are provided below, with corresponding revisions highlighted in track changes in the resubmitted files.

Comment 2: Line 63. Please include in the introduction the most significant/ representative anti-enzymatic effects.

Response 2: In response to the reviewer’s suggestion, we have specified the most significant anti-enzymatic effects, including tyrosinase, acetylcholinesterase and α-amylase inhibition, in the Introduction section of the revised manuscript (page 2, second paragraph, lines 66-67).

Comment 3: Line 224: “P ≤ 0.05”, “p” must appear in lower case letter. Same in line 332 and others.

Response 3: In response to the reviewer’s suggestion, we have corrected probability values as lowercase “p”, throughout the revised manuscript.

Comment 4: Line 228: “p-coumaric”, “p” must appear in italics. In addition, all the phytochemicals must be initialized with capital letters, i.e. Gallic acid. Change the use of capital letters in phytochemicals along the text.

Response 4: Thank you for your valuable feedback regarding the formatting of phytochemical names in the manuscript. In response to your suggestion, we have italicized the “p” in “p-coumaric” (page 7, first paragraph, line 244) to align with standard chemical nomenclature.

Regarding the capitalization of phytochemical names, we consulted the ACS Style Guide (3rd edition, 2006), which is commonly followed by MDPI journals, including Plants. The guide recommends using lowercase for common chemical names, such as “gallic acid,” unless they are proper nouns or trademarks. To ensure compliance with journal standards, we also contacted the MDPI editorial office, which confirmed that the current lowercase formatting of phytochemical names in our manuscript meets their style requirements and may remain unchanged. We greatly appreciate your attention to detail and are happy to make further adjustments if additional journal-specific preferences are identified.

Comment 5: In conclusion, in accordance to this reviewer criteria the manuscript in able to be published in the Plants Journal after minor modifications.

Response 5: We thank the reviewer for their positive evaluation and recommendation for publication in the journal Plants.

Reviewer 2 Report

Comments and Suggestions for Authors

Title: In Vitro Shoot Cultures of Micromeria graeca: Micropropagation and Evaluation of Methanolic Extracts for Anticancer and Antimicrobial Activity.

I recommended that manuscript could be accepted after MINOR MODIFICATIONS, based on:

ABSTRACT

  • Include name of author that described the plant the first time that is mentioned. Family (almost between parenthesis) could be positive.
  • First sentence should be divided in two.
  • Include quantitative results that complemented the qualitative described results.

INTRODUCTION

  • Family name is not in cursive and common name of plant is not in capital letter.

MATERIALS AND METHODS

  • Specify primary source of nodal segments of M. graeca with complete data (voucher number, collection date and geographical area).

GENERAL COMMENTS

  • Carefully review abbreviation use.
  • All mentioned plants as examples, should be included the name of author that described the first time that are mentioned.

Author Response

Response to Reviewer 2 Comments

We sincerely thank the reviewer for their valuable feedback and insightful comments on our manuscript titled “In Vitro Shoot Cultures of Micromeria graeca: Micropropagation and Evaluation of Methanolic Extracts for Anticancer and Antimicrobial Activity”. The reviewer’s suggestions have significantly improved the quality and clarity of our work. Below, we address each comment point by point, detailing the revisions made in response to the feedback. Changes in the revised manuscript are highlighted in track changes for clarity.

ABSTRACT

Comment 1: Include name of author that described the plant the first time that is mentioned. Family (almost between parenthesis) could be positive.

Response 1: Thank you for your suggestion. We have added the name of the author who first described the plant (Carl Linnaeus) at its initial mention and included the plant family in parentheses, as recommended. These changes are highlighted in the revised manuscript (page 1, first paragraph, line 18).

Comment 2: First sentence should be divided in two.

Response 2: Thank you for the suggestion. We have divided the first sentence into two as requested, with changes highlighted in track changes in the revised manuscript (page 1, first paragraph, line 20). This revision has greatly enhanced the text’s readability.

Comment 3: Include quantitative results that complemented the qualitative described results.

Response 3: Thank you for highlighting the need for quantitative data to complement the qualitative results. To address this, we have incorporated specific quantitative results, including fold-increase in rosmarinic acid content and IC50 values of CV test, into the Abstract to enhance the manuscript’s rigor and clarity. These additions are located on page 1, first paragraph, lines 33 [“(up to 4-fold)”] and 36 [“(IC50 values of CV assay ranging from 193 to 274 µg mL-1)”], and are highlighted in track changes in the revised manuscript.

INTRODUCTION

Comment 4: Family name is not in cursive and common name of plant is not in capital letter.

Response 4: Thank you for your feedback on the formatting of botanical names. We have ensured that the family name “Lamiaceae” is presented in roman type (not italics), consistent with the International Code of Nomenclature (ICN) and Plants Journal’s style guidelines. These changes are highlighted in track changes in the revised manuscript, particularly in the Introduction (page 1, first paragraph, line 18), and elsewhere (pages 2 (line 46), 3 (line 128), 6 (line 202), 7 (line 250)) in the revised manuscript.

Similarly, the common name “Greek savory or hyssop” is formatted in lowercase, except for the proper noun “Greek,” as per standard botanical conventions. If the journal prefers italicized family names or capitalized common names, please clarify, and we will adjust accordingly.

MATERIALS AND METHODS

Comment 5: Specify primary source of nodal segments of M. graeca with complete data (voucher number, collection date and geographical area).

Response 5: Thank you for your comment regarding the primary source of nodal segments of M. graeca. In the revised manuscript, we have included the complete details, including the voucher number, collection date, and geographical area, as requested. These details can now be found in the Materials and Methods section (page 15, paragraph 3 (subsection 3.1.1.), lines 435–438):

“Namely, aerial parts of M. graeca were randomly collected from natural population in Longos (Paxos, Greece), at latitude 39°13'41.93"N and longitude 20°9'40.37"E in July 2011. Voucher specimen (No. 14623) was deposited in the Herbarium collection of the Faculty of Sciences and Mathematics, University of Niš (HMN).”

We appreciate your attention to this detail and are happy to provide further clarification if needed.

GENERAL COMMENTS

Comment 6: Carefully review abbreviation use.

Response 6: Thank you for your feedback on abbreviation use. We have carefully reviewed and revised the abbreviations in the manuscript, ensuring consistency and clarity. Specific changes include modifications to existing abbreviations (e.g., Mg- and Mg+ now use singular forms instead of plural) and the introduction of new ones (AC-, AC+, Mg), all highlighted in track changes (see revised manuscript, page 20, lines 662, 663, 667 (new abbreviations); 668, 669 (modified abbreviations)). In line with modifying the existing abbreviations Mg- and Mg+, we slightly modified the text (singular, instead of plural) in subsection 2.4 of the Results and Discussion section (page 11, paragraph 4, lines 335–337).

Additionally, the existing abbreviation AC- is now first mentioned and defined in the text of the Results and Discussion section (page 3, paragraph 3, line 122) of the revised manuscript, whereas the existing abbreviation AC+ is now first mentioned and defined in the text of the Results and Discussion section (page 4, paragraph 1, line 155).

Comment 7: All mentioned plants as examples, should be included the name of author that described the first time that are mentioned.

Response 7: Thank you for your suggestion. We have added the names of the authors who first described each plant species at their first mention in the text. The changes to the text are highlighted in track changes in the revised manuscript, and are as follows:

Micromeria croatica (Pers.) Schott (page 3, paragraph 3, line 127)

Micromeria pulegium (Rochel) Benth. (page 3, paragraph 3, line 128)

Sideritis scardica Griseb. (page 4, line 152)

Micromeria thymifolia (Scop.) Fritsch (page 7, paragraph 1, line 247)

Micromeria juliana (L.) Benth. ex Rchb. (page 7, paragraph 1, line 248)

Micromeria myrtifolia Boiss. & Hohen. (page 7, paragraph 1, line 255)

Micromeria fruticosa (L.) Druce (page 11, paragraph 2, line 309)

Micromeria nervosa (Desf.) Benth. (page 11, paragraph 3, line 320)

Reviewer 3 Report

Comments and Suggestions for Authors

This study demonstrates a simple and straightforward in vitro culture protocol for Micromeria graeca and evaluates the use of activated charcoal not only on multiplication and biomass production but also its effect on the phenolic composition of cultures grown with or without activated charcoal. Additionally, and interestingly, antimicrobial and anticancer activities were also investigated. I suggest a few points for the authors’ consideration that would enhance the reproducibility of the protocol.

Line 15: There is no need to underline the abbreviation.

Lines 17-20: This sentence is too long. Consider ending it at “properties” and starting a new, shorter sentence about plant tissue culture before introducing the use of charcoal.

Line 39: Avoid using words that are already included in the title

Line 66: Consider adding a sentence on the importance of micropropagation protocols as a support for long-term conservation. For you consideration “Moreover, the development of a micropropagation protocol is a crucial prerequisite that also provides an opportunity for long-term conservation efforts.” Suggested references: https://doi.org/10.1007/s11240-020-01846-x https://doi.org/10.17660/ActaHortic.2025.1421.16

Line 107: Was the initiation phase efficient? Please add a comment on the in vitro establishment.

Line 148: “During shorter subculture intervals” - add the number of days in parentheses.

Line 402: Although a reference has been provided for the initiation phase, it is important to briefly describe the procedure within the manuscript to facilitate protocol reproducibility. I will allow others to replicate the method without needing to consult additional sources.

Line 408: Please provide details regarding the subculture?

Line 412: Specify the growing conditions?

Line 417: Clarify how dry weight was determined…

Line 420: Were all the roots were preserved or were pruned?

Line 421: add more information on the substrate. Was it autoclaved before use?

Line 553: twice?

Author Response

Response to Reviewer 3 Comments

Comment 1:

This study demonstrates a simple and straightforward in vitro culture protocol for Micromeria graeca and evaluates the use of activated charcoal not only on multiplication and biomass production but also its effect on the phenolic composition of cultures grown with or without activated charcoal. Additionally, and interestingly, antimicrobial and anticancer activities were also investigated. I suggest a few points for the authors’ consideration that would enhance the reproducibility of the protocol.

Response 1: We sincerely thank the reviewer for their positive and constructive feedback on our study. We appreciate their acknowledgment of our work on activated charcoal’s effects and the investigation of antimicrobial and anticancer activities. The reviewer’s valuable suggestions for enhancing the protocol’s reproducibility have been carefully addressed in our revisions.

Comment 2: Line 15: There is no need to underline the abbreviation.

Response 2: We agree with this comment, and have revised the manuscript accordingly.

Comment 3: Lines 17-20: This sentence is too long. Consider ending it at “properties” and starting a new, shorter sentence about plant tissue culture before introducing the use of charcoal.

Response 3: Thank you for the suggestions. We have divided the first sentence into two sentences, with changes highlighted in track changes in the revised manuscript (page 1, first paragraph, line 20). This revision has greatly enhanced the text’s readability.

We have addressed your suggestion to include a brief sentence about plant tissue culture by inserting the following sentence: "Plant tissue culture facilitates controlled in vitro propagation to study plant growth and bioactive properties." (page 1, first paragraph, lines 22-23) before introducing the use of activated charcoal.

Comment 4: Line 39: Avoid using words that are already included in the title

Response 4: Thank you for your suggestion. We have removed the redundant words as requested and replaced them with new keywords (axillary bud induction; Greek savory) to enhance clarity and specificity. These changes are highlighted in track changes in the revised manuscript (page 1, lines 42–43).

Comment 5: Line 66: Consider adding a sentence on the importance of micropropagation protocols as a support for long-term conservation. For you consideration “Moreover, the development of a micropropagation protocol is a crucial prerequisite that also provides an opportunity for long-term conservation efforts.” Suggested references: https://doi.org/10.1007/s11240-020-01846-x https://doi.org/10.17660/ActaHortic.2025.1421.16

Response 5: Thank you for suggesting the addition of a sentence about the importance of micropropagation protocols as a support for long-term conservation. We have incorporated the following sentence as recommended: "Developing micropropagation protocols is essential for in vitro plant studies and supports long-term conservation efforts." (page 2, paragraph 3, lines 70-71), supporting it by the suggested reference Volk et al. (2025), now listed in the References section.

Comment 6: Line 107: Was the initiation phase efficient? Please add a comment on the in vitro establishment.

Response 6: Yes, the initiation phase was quite efficient, given the collection season (late summer, instead of early spring).

As for your suggestion to include a comment on in vitro establishment, upon review, we noted that the original manuscript already contains a sentence addressing this in Section 2.1: "Proliferating shoot cultures of M. graeca were successfully established from surface-sterilized one-node stem segments (~1 cm) of wild-growing plants, with light green axillary buds emerging within 7–10 days after inoculation on basal MS medium [37]." To further highlight this information and improve clarity, we have added a new subsection title, "2.1.1. Shoot Culture Establishment" (page 3, line 112), within Section 2.1. Micropropagation. Additionally, upon your request, shoot culture establishment is now described in details in the section Materials and Methods (page 15, paragraphs 3 and 4, lines 435–450; page 16, paragraph 1, lines 451–456) of the revised manuscript. We believe this addresses your suggestion while avoiding redundancy.

Comment 7: Line 148: “During shorter subculture intervals” - add the number of days in parentheses.

Response 7: The number of days is now added in parentheses as requested, in the revised version of the manuscript (page 4, paragraph 1, line 156).

Comment 8: Line 402: Although a reference has been provided for the initiation phase, it is important to briefly describe the procedure within the manuscript to facilitate protocol reproducibility. I will allow others to replicate the method without needing to consult additional sources.

Response 8: Thank you for your suggestion to briefly describe the initiation phase procedure within the manuscript to enhance protocol reproducibility. We have revised subsection 3.1.1 (now entitled “Plant material, shoot culture establishment and maintenance”, page 15, line 433) to include a concise description of the initiation phase, ensuring that readers can replicate the method without consulting additional sources. The changes are highlighted in track changes in the revised manuscript (page 15, paragraphs 3 and 4; page 16, paragraph 1).

Comment 9: Line 408: Please provide details regarding the subculture?

Response 9: We have now provided details regarding subculture (“The stock shoot cultures were maintained on basal MS medium, under previously described growth conditions, and subcultured on fresh medium every 6 weeks.”). The changes are highlighted in track changes in the revised manuscript (page 16, paragraph 1, lines 454–456).

Comment 10: Line 412: Specify the growing conditions?

Response 10: We have specified the growing conditions as “Cultures were maintained in a growth chamber under a 16‑h photoperiod, with a photon flux density of 45 µmol m‑2 s‑1 provided by cool white fluorescent tubes, at 25 ± 2 °C.” (page 15, paragraph 4, lines 448–450). As for the stock cultures maintenance and the shoot multiplication experiments, we refer to these very conditions in the form of: “(maintained...) under previously described growth conditions” (page 16, paragraph 1, lines 455–456; page 16, paragraph 2, line 466)

Comment 11: Line 417: Clarify how dry weight was determined…

Response 11: We have specified both fresh and dry weight measurements, by adding “For weight measurements, the aerial parts of the plants were used. Fresh weight was recorded upon removal from the culture (on the day of sampling), whereas dry weight was recorded after drying the explants at 60 °C for 72 h.” in Materials and Methods section. The changes are highlighted in track changes in the revised manuscript (page 16, paragraph 3, lines 470–472).

Comment 12: Line 420: Were all the roots were preserved or were pruned?

Response 12: In the acclimatization experiment, our intention was to preserve all the roots. They were only thoroughly washed to remove any adherent medium before planting, so it is possible that during that procedure some of them may have been damaged or removed, unintentionally.

Comment 13: Line 421: add more information on the substrate. Was it autoclaved before use?

Response 13: No, the substrate was not autoclaved before use.

Comment 14: Line 553: twice?

Response 14: Yes, twice, as the experiment was repeated.     
